# Learning convex polytopes with margin

**Lee-Ad Gottlieb**
Ariel University
leead@ariel.ac.il

**Eran Kaufman**
Ariel University
erankfmn@gmail.com

**Aryeh Kontorovich**
Ben-Gurion University
karyeh@bgu.sc.il

**Gabriel Nivasch**
Ariel University
gabrieln@ariel.ac.il

## Abstract

We present an improved algorithm for properly learning convex polytopes in the realizable PAC setting from data with a margin. Our learning algorithm constructs a consistent polytope as an intersection of about $t \log t$ halfspaces with margins in time polynomial in $t$ (where $t$ is the number of halfspaces forming an optimal polytope).

We also identify distinct generalizations of the notion of margin from hyperplanes to polytopes and investigate how they relate geometrically; this result may be of interest beyond the learning setting.

## 1   Introduction

In the theoretical PAC learning setting [Valiant, 1984], one considers an abstract *instance space* $\mathcal{X}$ — which, most commonly, is either the Boolean cube $\{0,1\}^d$ or the Euclidean space $\mathbb{R}^d$. For the former setting, an extensive literature has explored the statistical and computational aspects of learning Boolean functions [Angluin, 1992, Hellerstein and Servedio, 2007]. Yet for the Euclidean setting, a corresponding theory of learning geometric concepts is still being actively developed [Kwek and Pitt, 1998, Jain and Kinber, 2003, Anderson et al., 2013, Kane et al., 2013]. The focus of this paper is the latter setting.

The simplest nontrivial geometric concept is perhaps the halfspace. These concepts are well-known to be hard to agnostically learn [Höffgen et al., 1995] or even approximate [Amaldi and Kann, 1995, 1998, Ben-David et al., 2003]. Even the realizable case, while commonly described as "solved" via the Perceptron algorithm or linear programming (LP), is not straightforward: The Perceptron's runtime is quadratic in the inverse-margin, while solving the consistent hyperplane problem in strongly polynomial time is equivalent to solving the general LP problem in strongly polynomial time [Nikolov, 2018, Chvátal], a question that has been open for decades [Bárász and Vempala, 2010]. Thus, an unconditional (i.e., infinite-precision and independent of data configuration in space) polynomial-time solution for the consistent hyperplane problem hinges on the strongly polynomial LP conjecture.

If we consider not a single halfspace, but polytopes defined by the intersection of multiple halfspaces, the computational and generalization bounds rapidly become more pessimistic. Megiddo [1988] showed that the problem of deciding whether two sets of points in general space can be separated by the intersection of two hyperplanes is NP-complete, and Khot and Saket [2011] showed that "unless NP = RP, it is hard to (even) weakly PAC-learn intersection of two halfspaces", even when allowed the richer class of $O(1)$ intersecting halfspaces. Under cryptographic assumptions, Klivans and Sherstov [2009] showed that learning an intersection of $n^\varepsilon$ halfspaces is intractable regardless of hypothesis representation.

Since the margin assumption is what allows one to find a consistent hyperplane in provably strongly polynomial time, it is natural to seek to generalize this scheme to intersections of $t$ halfspaces each with margin $\gamma$; we call this the $\gamma$-*margin* of a $t$-polytope. This problem was considered by Arriaga and Vempala [2006], who showed that such a polytope can be learned (in dimension $d$) in time

$$O(dmt) + (t \log t)^{O((t/\gamma^2)\log(t/\gamma))}$$

with sample complexity $m = O\left((t/\gamma^2)\log(t)\log(t/\gamma)\right)$ (where we have taken the PAC-learning parameters $\varepsilon, \delta$ to be constants). In fact, they actually construct a candidate $t$-polytope as their learner; as such, their approach is an example of *proper learning*, where the hypothesis is chosen from the same concept class as the true concept. In contrast, Klivans and Servedio [2008] showed that a $\gamma$-margin $t$-polytope can be learned by constructing a function that approximates the polytope's behavior, without actually constructing a $t$-polytope. This is an example of *improper learning*, where the hypothesis is selected from a broader class than that of the true concept. They achieved a runtime of

$$\min\left\{ d(t/\gamma)^{O(t \log t \log(1/\gamma))}, d\left(\frac{\log t}{\gamma}\right)^{O(\sqrt{1/\gamma}\log t)} \right\}$$

and sample complexity $m = O\left((1/\gamma)^{t \log t \log(1/\gamma)}\right)$. Very recently, Goel and Klivans [2018] improved on this latter result, constructing a function hypothesis in time $\text{poly}(d, t^{O(1/\gamma)})$, with sample complexity exponential in $\gamma^{-1/2}$.

**Our results.**    The central contribution of the paper is improved algorithmic runtimes and sample complexity for computing separating polytopes (Theorem 7). In contrast to the algorithm of Arriaga and Vempala [2006], whose runtime is exponential in $t/\gamma^2$, and to that of [Goel and Klivans, 2018], whose sample complexity is exponential in $\gamma^{-1/2}$, we give an algorithm with polynomial sample complexity $m = \tilde{O}(t/\gamma^2)$ and runtime only $m^{\tilde{O}(1/\gamma^2)}$. We accomplish this by constructing an $O(t \log m)$-polytope that correctly separates the data. This means that our hypothesis is drawn from a broader class than the $t$-polytopes of Arriaga and Vempala [2006] (allowing faster runtime), but from a much narrower class than the functions of Klivans and Servedio [2008], Goel and Klivans [2018] (allowing for improved sample complexity).

Complementing our algorithm, we provide the first nearly matching hardness-of-approximation bounds, which demonstrate that an exponential dependence on $t\gamma^{-2}$ is unavoidable for the computation of separating $t$-polytopes, under standard complexity-theoretic assumptions (Theorem 6). This motivates our consideration of $O(t \log m)$-polytopes instead.

Our final contribution is in introducing a new and intuitive notion of polytope margin: This is the $\gamma$-*envelope* of a convex polytope, defined as all points within distance $\gamma$ of the polytope boundary, as opposed to the above $\gamma$-*margin* of the polytope, defined as the intersection of the $\gamma$-margins of the hyperplanes forming the polytope. (See Figure 2 for an illustration, and Section 2 for precise definitions.) Note that these two objects may exhibit vastly different behaviors, particularly at a sharp intersection of two or more hyperplanes. It seems to us that the envelope of a polytope is a more natural structure than its margin, yet we find the margin more amenable to the derivation of both VC-bounds (Lemma 1) and algorithms (Theorem 7). We demonstrate that results derived for margins can be adapted to apply to envelopes as well. We prove that when confined to the unit ball, the $\gamma$-envelope fully contains within it the $(\gamma^2/2)$-margin (Theorem 10), and this implies that statistical and algorithmic results for the latter hold for the former as well.

**Related work.**    When general convex bodies are considered under the uniform distribution[1] (over the unit ball or cube), exponential (in dimension and accuracy) sample-complexity bounds were obtained by Rademacher and Goyal [2009]. This may motivate the consideration of convex polytopes, and indeed a number of works have studied the problem of learning convex polytopes, including Hegedüs [1994], Kwek and Pitt [1998], Anderson et al. [2013], Kane et al. [2013], Kantchelian et al. [2014]. Hegedüs [1994] examines query-based exact identification of convex polytopes with integer vertices, with runtime polynomial in the number of vertices (note that the number of vertices

can be exponential in the number of facets [Matoušek, 2002]). Kwek and Pitt [1998] also rely on membership queries (see also references therein regarding prior results, as well as strong positive results in 2 dimensions). Anderson et al. [2013] efficiently approximately recover an unknown simplex from uniform samples inside it. Kane et al. [2013] learn halfspaces under the log-concave distributional assumption.

The recent work of Kantchelian et al. [2014] bears a superficial resemblance to ours, but the two are actually not directly comparable. What they term *worst case margin* will indeed correspond to our *margin*. However, their optimization problem is non-convex, and the solution relies on heuristics without rigorous run-time guarantees. Their generalization bounds exhibit a better dependence on the number $t$ of halfspaces than our Lemma 3 ($O(\sqrt{t})$ vs. our $O(t \log t)$). However, the hinge loss appearing in their Rademacher-based bound could be significantly worse than the 0-1 error appearing in our VC-based bound. We stress, however, that the main contribution of our paper is algorithmic rather than statistical.

## 2 Preliminaries

**Notation.** For $\mathbf{x} \in \mathbb{R}^d$, we denote its Euclidean norm $\|\mathbf{x}\|_2 := \sqrt{\sum_{i=1}^d \mathbf{x}(i)^2}$ by $\|\mathbf{x}\|$ and for $n \in \mathbb{N}$, we write $[n] := \{1, \dots, n\}$. Our **instance space** $\mathcal{X}$ is the unit ball in $\mathbb{R}^d$: $\mathcal{X} = \{\mathbf{x} \in \mathbb{R}^d : \|\mathbf{x}\| \leq 1\}$. We assume familiarity with the notion of VC-dimension as well as with basic PAC definitions such as *generalization error* (see, e.g., Kearns and Vazirani [1997]).

**Polytopes.** A (convex) polytope $P \subset \mathbb{R}^d$ is the convex hull of finitely many points: $P = \mathrm{conv}(\{\mathbf{x}_1, \dots, \mathbf{x}_n\})$. Alternatively, it can be defined by $t$ hyperplanes $(\mathbf{w}_i, b_i) \in \mathbb{R}^d \times \mathbb{R}$ where $\|\mathbf{w}_i\| = 1$ for each $i$:

$$P = \left\{ \mathbf{x} \in \mathbb{R}^d : \min_{i \in [t]} \mathbf{w}_i \cdot \mathbf{x} + b_i \geq 0 \right\}. \tag{1}$$

A hyperplane $(\mathbf{w}, b)$ is said to classify a point $\mathbf{x}$ as positive (resp., negative) with margin $\gamma$ if $\mathbf{w} \cdot \mathbf{x} + b \geq \gamma$ (resp., $\leq -\gamma$). Since $\|\mathbf{w}\| = 1$, this means that $\mathbf{x}$ is $\gamma$-far from the hyperplane $\{\mathbf{x}' \in \mathbb{R}^d : \mathbf{w} \cdot \mathbf{x}' + b = 0\}$, in $\ell_2$ distance.

**Margins and envelopes.** We consider two natural ways of extending this notion to polytopes: the $\gamma$-margin and the $\gamma$-envelope. For a polytope defined by $t$ hyperplanes as in (1), we say that $\mathbf{x}$ is in the *inner $\gamma$-margin* of $P$ if

$$0 \leq \min_{i \in [t]} \mathbf{w}_i \cdot \mathbf{x} + b_i \leq \gamma$$

and that $\mathbf{x}$ is in the *outer $\gamma$-margin* of $P$ if

$$0 \geq \min_{i \in [t]} \mathbf{w}_i \cdot \mathbf{x} + b_i \geq -\gamma.$$

Similarly, we say that $\mathbf{x}$ is in the *outer $\gamma$-envelope* of $P$ if $\mathbf{x} \notin P$ and $\inf_{\mathbf{p} \in P} \|\mathbf{x} - \mathbf{p}\| \leq \gamma$ and that $\mathbf{x}$ is in the *inner $\gamma$-envelope* of $P$ if $\mathbf{x} \in P$ and $\inf_{\mathbf{p} \notin P} \|\mathbf{x} - \mathbf{p}\| \leq \gamma$.

We call the union of the inner and the outer $\gamma$-margins the *$\gamma$-margin*, and we denote it by $\partial P^{[\gamma]}$. Similarly, we call the union of the inner and the outer $\gamma$-envelopes the *$\gamma$-envelope*, and we denote it by $\partial P^{(\gamma)}$.

The two notions are illustrated in Figure 2. As we show in Section 4 below, the inner envelope coincides with the inner margin, but this is not the case for the outer objects: The outer margin always contains the outer envelope, and could be of arbitrarily larger volume.

**Fat hyperplanes and polytopes.** Binary classification requires a collection of *concepts* mapping the instance space (in our case, the unit ball in $\mathbb{R}^d$) to $\{-1, 1\}$. However, given a hyperplane $(\mathbf{w}, b)$ and a margin $\gamma$, the function $f_{\mathbf{w}, b} : \mathbb{R}^d \to \mathbb{R}$ given by $f_{\mathbf{w}, b}(\mathbf{x}) = \mathbf{w} \cdot \mathbf{x} + b$ partitions $\mathbb{R}^d$ into three regions: *positive* $\{\mathbf{x} \in \mathbb{R}^d : f_{\mathbf{w}, b}(\mathbf{x}) \geq \gamma\}$, *negative* $\{\mathbf{x} \in \mathbb{R}^d : f_{\mathbf{w}, b}(\mathbf{x}) \leq -\gamma\}$, and *ambiguous* $\{\mathbf{x} \in \mathbb{R}^d : |f_{\mathbf{w}, b}(\mathbf{x})| < \gamma\}$. We use a standard device (see, e.g., Hanneke and Kontorovich [2017,

Section 4]) of defining an auxiliary instance space $\mathcal{X}' = \mathcal{X} \times \{-1, 1\}$ together with the concept class $\mathcal{H}_\gamma = \big\{h_{\mathbf{w},b} : \mathbf{w} \in \mathbb{R}^d, b \in \mathbb{R}, \|\mathbf{w}\| = 1/\gamma\big\}$, where, for all $(\mathbf{x}, y) \in \mathcal{X}'$,

$$h_{\mathbf{w},b}(\mathbf{x}, y) = \begin{cases} \text{sign}(y(\mathbf{w} \cdot x + b)), & |\mathbf{w} \cdot \mathbf{x} + b| \geq \gamma \\ -1, & \text{else.} \end{cases}$$

It is shown in [Hanneke and Kontorovich, 2017, Lemma 6] that[2]

**Lemma 1.** *The VC-dimension of $\mathcal{H}_\gamma$ is at most $(2/\gamma + 1)^2$.*

Analogously, we define the concept class $\mathcal{P}_{t,\gamma}$ of $\gamma$-fat $t$-polytopes as follows. Each $h_P \in \mathcal{P}_{t,\gamma}$ is induced by some $t$-halfspace intersection $P$ as in (1). The label of a pair $(\mathbf{x}, y) \in \mathcal{X}'$ is determined as follows: If $\mathbf{x}$ is in the $\gamma$-margin of $P$, then the pair is labeled $-1$ irrespective of $y$. Otherwise, if $\mathbf{x} \in P$ and $y = 1$, or else $\mathbf{x} \notin P$ and $y = -1$, then the pair is labeled $1$. Otherwise, the pair is labeled $-1$.

**Lemma 2.** *The VC-dimension of $\mathcal{P}_{t,\gamma}$ in $d$ dimensions is at most*

$$\min \left\{ 2(d+1)t \log(3t), 2vt \log(3t) \right\},$$

*where $v = (2/\gamma + 1)^2$.*

*Proof.* The family of intersections of $t$ concept classes of VC-dimension at most $v$ is bounded by $2vt \log(3t)$ [Blumer et al., 1989, Lemma 3.2.3]. Since the class of $d$-dimensional hyperplanes has VC-dimension $d + 1$ [Long and Warmuth, 1994], the family of polytopes has VC-dimension at most $2(d+1)t \log(3t)$. The second part of the bound is obtained by applying Blumer et al. [1989, Lemma 3.2.3] to the VC bound in Lemma 1.

$\square$

**Generalization bounds.**   The following VC-based generalization bounds are well-known; the first one may be found in, e.g., Cristianini and Shawe-Taylor [2000], while the second one in Anthony and Bartlett [1999].

**Lemma 3.** *Let $H$ be a class of learners with VC-dimension $d_{\text{VC}}$. If a learner $h \in H$ is consistent on a random sample $S$ of size $m$, then with probability at least $1 - \delta$ its generalization error is*

$$\text{err}(h) \leq \frac{2}{m}\big(d_{\text{VC}} \log(2em/d_{\text{VC}}) + \log(2/\delta)\big).$$

**Dimension reduction.**   The Johnson-Lindenstrauss (JL) transform [Johnson and Lindenstrauss, 1982] takes a set $S$ of $m$ vectors in $\mathbb{R}^d$ and projects them into $k = O(\varepsilon^{-2} \log m)$ dimensions, while preserving all inter-point distances and vector norms up to $1 + \varepsilon$ distortion. That is, if $f : \mathbb{R}^d \to \mathbb{R}^k$ is a linear embedding realizing the guarantees of the JL transform on $S$, then for every $\mathbf{x} \in S$ we have

$$(1 - \varepsilon)\|\mathbf{x}\| \leq \|f(\mathbf{x})\| \leq (1 + \varepsilon)\|\mathbf{x}\|,$$

and for every $\mathbf{x}, \mathbf{y} \in S$ we have

$$(1 - \varepsilon)\|\mathbf{x} - \mathbf{y}\| \leq \|f(\mathbf{x} - \mathbf{y})\| \leq (1 + \varepsilon)\|\mathbf{x} - \mathbf{y}\|.$$

The JL transform can be realized with probability $1 - n^{-c}$ for any constant $c \geq 1$ by a randomized linear embedding, for example a projection matrix with entries drawn from a normal distribution [Achlioptas, 2003]. This embedding is *oblivious*, in the sense that the matrix can be chosen without knowledge of the set $S$.

It is an easy matter to show that the JL transform can also be used to approximately preserve distances to hyperplanes, as in the following lemma.

**Lemma 4.** *Let $S$ be set of $d$-dimensional vectors in the unit ball, $T$ be a set of normalized vectors, and $f : \mathbb{R}^d \to \mathbb{R}^k$ a linear embedding realizing the guarantees of the JL transform. Then for any $0 < \varepsilon < 1$ and some $k = O((\log |S \cup T|)/\varepsilon^2)$, with probability $1 - |S \cup T|^{-c}$ (for any constant $c > 1$) we have for all $\mathbf{x} \in S$ and $\mathbf{t} \in T$ that*

$$f(\mathbf{t}) \cdot f(\mathbf{x}) \in \mathbf{t} \cdot \mathbf{x} \pm \varepsilon.$$

*Proof.* Let the constant in $k$ be chosen so that the JL transform preserves distances and norms among $S \cup T$ within a factor $1 + \varepsilon'$ for $\varepsilon' = \varepsilon/5$. By the guarantees of the JL transform for the chosen value of $k$, we have that

$$
\begin{aligned}
f(\mathbf{t}) \cdot f(\mathbf{x}) &= \frac{1}{2}\left[\|f(\mathbf{t})\|^2 + \|f(\mathbf{x})\|^2 - \|f(\mathbf{t}) - f(\mathbf{x})\|^2\right] \\
&\leq \frac{1}{2}\left[(1 + \varepsilon')^2(\|\mathbf{t}\|^2 + \|\mathbf{x}\|^2) - (1 - \varepsilon')^2\|\mathbf{t} - \mathbf{x}\|^2\right] \\
&< \frac{1}{2}\left[(1 + 3\varepsilon')(\|\mathbf{t}\|^2 + \|\mathbf{x}\|^2) - (1 - 2\varepsilon')\|\mathbf{t} - \mathbf{x}\|^2\right] \\
&< \frac{1}{2}\left[5\varepsilon'(\|\mathbf{t}\|^2 + \|\mathbf{x}\|^2) + \mathbf{t} \cdot \mathbf{x}\right] \\
&\leq 5\varepsilon' + \mathbf{t} \cdot \mathbf{x}. \\
&= \varepsilon + \mathbf{t} \cdot \mathbf{x}.
\end{aligned}
$$

A similar argument gives that $f(\mathbf{t}) \cdot f(\mathbf{x}) > -\varepsilon + \mathbf{t} \cdot \mathbf{x}$. $\qquad\square$

## 3 Computing and learning separating polytopes

In this section, we present algorithms to compute and learn $\gamma$-fat $t$-polytopes. We begin with hardness results for this problem, and show that these hardness results justify algorithms with run time exponential in the dimension or the square of the reciprocal of the margin. We then present our algorithms.

### 3.1 Hardness

We show that computing separating polytopes is NP-hard, and even hard to approximate. We begin with the case of a single hyperplane. The following preliminary lemma builds upon Amaldi and Kann [1995, Theorem 10].

**Lemma 5.** *Given a labelled point set $S$ ($n = |S|$) with $p$ negative points, let $h^*$ be a hyperplane that places all positive points of $S$ on its positive side, and maximizes the number of negative points on its negative size — let* opt *be the number of these negative points. Then it is* NP-*hard to find a hyperplane $\tilde{h}$ consistent with all positive points, and which places at least* $\text{opt}/p^{1-o(1)}$ *negative points on on the negative side of $\tilde{h}$. This holds even when the optimal hyperplane correctly classifying* opt *points has margin $\gamma \geq \frac{1}{4\sqrt{\text{opt}}}$.*

*Proof.* We reduce from maximum independent set, which for $p$ vertices is hard to approximate to within $p^{1-o(1)}$ [Zuckerman, 2007]. Given a graph $G = (V, E)$, for each vetex $v_i \in V$ place a negative point on the basis vector $\mathbf{e}_i$. Now place a positive point at the origin, and for each edge $(v_i, v_j) \in E$, place a positive point at $(\mathbf{e}_i + \mathbf{e}_j)/2$.

Consider a hyperplane consistent with the positive points and placing opt negative points on the negative side: These negative points must represent an independent set in $G$, for if $(v_i, v_j) \in E$, then by construction the midpoint of $\mathbf{e}_i, \mathbf{e}_j$ is positive, and so both $\mathbf{e}_i, \mathbf{e}_j$ cannot lie on the negative side of the hyperplane.

Likewise, if $G$ contained an independent set $V' \subset V$ of size opt, then we consider the hyperplane defined by the equation $\mathbf{w} \cdot \mathbf{x} + \frac{3}{4\sqrt{\text{opt}}} = 0$, where coordinate $\mathbf{w}(j) = -\frac{1}{\sqrt{\text{opt}}}$ if $v_j \in V'$ and $\mathbf{w}(j) = 0$ otherwise. It is easily verified that the distance from the hyperplane to a negative point (i.e. a basis vector) is $-\frac{1}{\sqrt{\text{opt}}} + \frac{3}{4\sqrt{\text{opt}}} = -\frac{1}{4\sqrt{\text{opt}}}$, to the origin is $\frac{3}{4\sqrt{\text{opt}}}$, and to all other positive points is at least $-\frac{1}{2\sqrt{\text{opt}}} + \frac{3}{4\sqrt{\text{opt}}} = \frac{1}{4\sqrt{\text{opt}}}$. $\qquad\square$

We can now extend the above result for a hyperplane to similar ones for polytopes:

**Theorem 6.** *Given a labelled point set $S$ ($n = |S|$) with $p$ negative points, let $H^*$ be a collection of $t$ halfspaces whose intersection partitions $S$ into positive and negative sets. Then it is* NP*-hard to find a collection $\tilde{H}$ of size less than $tp^{1-o(1)}$ whose intersection also partitions $S$ into positive and negative sets. This holds even when all hyperplanes have margin $\gamma \geq \frac{1}{4\sqrt{p/t}}$,*

*Proof.* The reduction is from minimum coloring, which is hard to approximate within a factor of $n^{1-o(1)}$ [Zuckerman, 2007]. The construction is identical to that of the proof of Lemma 5. In particular, a set of vertices in $G$ assigned the same color necessarily form an independent set, and so their corresponding negative points in $S$ can be separated from all positive points by some halfspace, and vice-versa.

The only difficulty in the reduction is our insistence that the margin must be of size at least $\frac{1}{4\sqrt{p/t}}$; as in Lemma 5, this holds only when the halfspaces are restricted to separate at most $\mathrm{opt} = p/t$ points. But there is no guarantee that the optimal coloring satisfies this requirement, that is if the optimal coloring possesses $t$ colors, that each color represents only $p/t$ vertices. To this end, if a color in the optimal $t$-coloring of $G$ covers more than $p/t$ vertices, we partition it into a set of colors, each coloring no more than $p/t$ vertices. This increases the total number of colors to at most $2t$, which does not affect the hardness-of-approximation result. $\qquad\square$

The Exponential Time Hypothesis (ETH) posits that maximum independent set and minimum coloring cannot be solved in less than $c^n$ operations (for some constant $c$)[3]. As Lemma 5 asserts that the separating hyperplane problem remains hard for margin $\gamma \geq \frac{1}{4\sqrt{\mathrm{opt}}} \geq \frac{1}{4\sqrt{p}}$, we cannot hope to find an optimal solution in time less than $c^p \geq c^{1/(16\gamma^2)}$. Likewise, as Theorem 6 asserts that the separating $t$-polytope problem remains hard for margin $\gamma \geq \frac{1}{4\sqrt{p/t}}$ we cannot hope to find a consistent $t$-polytope in time less than $c^p \geq c^{t/(16\gamma^2)}$. This justifies the exponential dependence on $t\gamma^{-2}$ in the algorithm of Arriaga and Vempala [2006], and implies that to avoid an exponential dependence on $t$ in the runtime, we should consider broader hypothesis class, for example $O(t \log m)$-polytopes.

## 3.2 Algorithms

Here we present algorithms for computing polytopes, and use them to give an efficient algorithm for learning polytopes.

In what follows, we give two algorithms inspired by the work of Arriaga and Vempala [2006]. Both have runtime faster than the algorithm of Arriaga and Vempala [2006], and the second is only polynomial in $t$.

**Theorem 7.** *Given a labelled point set $S$ ($n = |S|$) for which some $\gamma$-fat $t$-polytope correctly separates the positive and negative points (i.e., the polytope is* consistent*), we can compute the following with high probability:*

 1. *A consistent $(\gamma/4)$-fat $t$-polytope in time $n^{O(t\gamma^{-2}\log(1/\gamma))}$.*

 2. *A consistent $(\gamma/4)$-fat $O(t \log n)$-polytope in time $n^{O(\gamma^{-2}\log(1/\gamma))}$.*

Before proving Theorem 7, we will need a preliminary lemma:

**Lemma 8.** *Given any $0 < \delta < 1$, there exists a set $V$ of unit vectors of size $|V| = \delta^{-O(d)}$ with the following property: For any unit vector $\mathbf{w}$, there exists a $\mathbf{v} \in V$ that satisfies $\mathbf{v} \cdot \mathbf{x} \in \mathbf{w} \cdot \mathbf{x} \pm \delta$ for all vectors $\mathbf{x}$ with $\|\mathbf{x}\| \leq 1$. The set $V$ can be constructed in time $\delta^{-O(d)}$ with high probability.*

This implies that if a set $S$ admits a hyperplane $(\mathbf{w}, b)$ with margin $\gamma$, then $S$ admits a hyperplane $(\mathbf{v}, b)$ (for $\mathbf{v} \in V$) with margin at least $\gamma - \delta$.

*Proof.* We take $V$ to be a $\delta$-net of the unit ball, a set satisfying that every point on the ball is within distance $\delta$ of some point in $V$. Then $|V| \leq (1 + 2/\delta)^d$ [Vershynin, 2010, Lemma 5.2]. For any unit vector $\mathbf{w}$ we have for some $\mathbf{v} \in V$ that $\|\mathbf{w} - \mathbf{v}\| \leq \delta$, and so for any vector $\mathbf{x}$ satisfying $\|\mathbf{x}\| \leq 1$ we have

$$|\mathbf{w} \cdot \mathbf{x} - \mathbf{v} \cdot \mathbf{x}| = |(\mathbf{w} - \mathbf{v}) \cdot \mathbf{x}| \leq \|\mathbf{w} - \mathbf{v}\| \leq \delta.$$

The net can be constructed by a randomized greedy algorithm. By coupon-collector analysis, it suffices to sample $O(|V| \log |V|)$ random unit vectors. For example, each can be chosen by sampling its coordinate from $N(0,1)$ (the standard normal distribution), and then normalizing the vector. The resulting set contains within it a $\delta$-net. $\square$

*Proof of Theorem 7.* We first apply the Johnson-Lindenstrauss transform to reduce dimension of the points in $S$ to $k = O(\gamma^{-2} \log(n + t)) = O(\gamma^{-2} \log n)$ while achieving the guarantees of Lemma 4 for the points of $S$ and the $t$ halfspaces forming the optimal $\gamma$-fat $t$-polytope, with parameter $\varepsilon = \frac{\gamma}{12}$. In the embedded space, we extract a $\delta$-net $V$ of Lemma 8 with parameter $\delta = \frac{\gamma}{12}$, and we have $|V| = \delta^{-O(k)}$. Now define the set $B$ consisting of all values of the form $\frac{\gamma i}{12}$ for integer $i = \{0, 1, \ldots, \lfloor 12/\gamma \rfloor\}$. It follows that for each $d$-dimensional halfspace $(\mathbf{w}, b)$ forming the original $\gamma$-fat $t$-polytope, there is a $k$-dimensional halfspace $(\mathbf{v}, b')$ with $\mathbf{v} \in V$ and $b' \in B$ satisfying $\mathbf{v} \cdot f(\mathbf{x}) + b' \in \mathbf{w} \cdot \mathbf{x} + b \pm \gamma/4$ for every $\mathbf{x} \in S$. Given $(\mathbf{v}, b')$, we can recover an approximation to $(\mathbf{w}, b)$ in the $d$-dimensional origin space thus: Let $S' \subset S$ include only those points $\mathbf{x} \in S$ for which $|\mathbf{v} \cdot f(\mathbf{x}) + b'| \geq \frac{3\gamma}{4}$, and it follows that $|\mathbf{w} \cdot \mathbf{x} + b| \geq \frac{3\gamma}{4} - \frac{\gamma}{4} = \frac{\gamma}{2}$. As $S'$ is a separable point set with margin $\Theta(\gamma)$, we can run the Perceptron algorithm on $S'$ in time $O(dn\gamma^{-2})$, and find a $d$-dimensional halfspace $\mathbf{w}'$ consistent with $\mathbf{w}$ on all points at distance $\frac{\gamma}{4}$ or more from $\mathbf{w}$. We will refer to $\mathbf{w}'$ as the $d$-dimensional mirror of $\mathbf{v}$.

We compute the $d$-dimensional mirror of every vector in $V$ for every candidate value in $B$. We then enumerate all possible $t$-polytopes by taking intersections of all combinations of $t$ mirror halfspaces, in total time

$$(1/\gamma)^{O(kt)} = n^{O(t\gamma^{-2} \log(1/\gamma))},$$

and choose the best one consistent with $S$. The first part of the theorem follows.

Better, we may give a greedy algorithm with a much improved runtime: First note that as the intersection of $t$ halfspaces correctly classifies all points, the best halfspace among them correctly classifies at least a $(1/t)$-fraction of the negative points with margin $\gamma$. Hence it suffices to find the $d$-dimensional mirror which is consistent with all positive points and maximizes the number of correct negative points, all with margin $\frac{\gamma}{4}$. We choose this halfspace, remove from $S$ the correctly classified negative points, and iteratively search for the next best halfspace. After $ct \log n$ iterations (for an appropriate constant $c$), the number of remaining points is

$$n(1 - \Omega(1/t))^{ct \log n} < ne^{-\ln n} = 1,$$

and the algorithm terminates. $\square$

Having given an algorithm to *compute* $\gamma$-fat $t$-polytopes, we can now give an efficient algorithm to *learn* $\gamma$-fat $t$-polytopes. We sample $m$ points, and use the second item of Theorem 7 to find a $(\gamma/4)$-fat $O(t \log m)$-polytope consistent with the sample. By Lemma 2, the class of polytopes has VC-dimension $O(\gamma^{-2} t \log m)$. The size of $m$ is chosen according to Lemma 3, and we conclude:

**Theorem 9.** *There exists an algorithm that learns $\gamma$-fat $t$-polytopes with sample complexity*

$$m = O\left(\frac{t}{\varepsilon \gamma^2} \log^2 \frac{t}{\varepsilon \gamma} + \log \frac{1}{\delta}\right)$$

*in time $m^{O((1/\gamma^2) \log(1/\gamma))}$, where $\varepsilon, \delta$ are the desired accurcy and confidence levels.*

## 4  Polytope margin and envelope

In this section, we show that the notions of margin and envelope defined in Section 2 are, in general, quite distinct. Fortunately, when confined to the unit ball $\mathcal{X}$, one can be used to approximate the other.

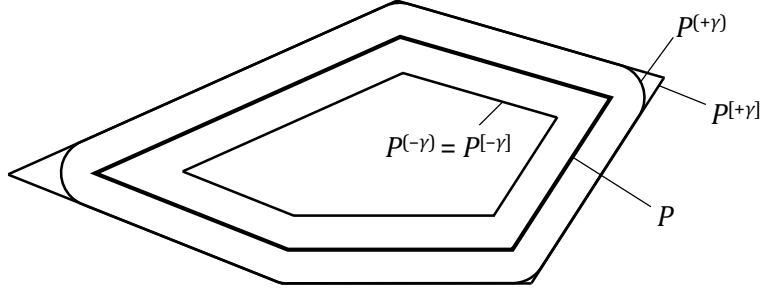

Figure 1: Expansion and contraction of a polytope by $\gamma$.

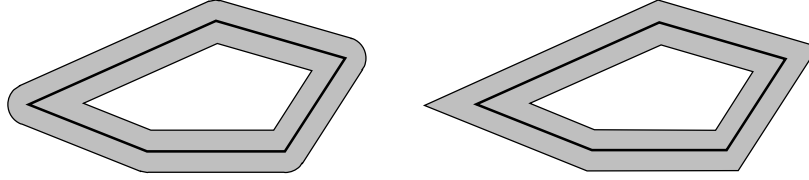

Figure 2: The $\gamma$-envelope $\partial P^{(\gamma)}$ (left) and $\gamma$-margin $\partial P^{[\gamma]}$ (right) of a polytope $P$.

Given two sets $S_1, S_2 \subseteq \mathbb{R}^d$, their *Minkowski sum* is given by $S_1 + S_2 = \{\mathbf{p} + \mathbf{q} : \mathbf{p} \in S_1, \mathbf{q} \in S_2\}$, and their *Minkowski difference* is given by $S_1 - S_2 = \{\mathbf{p} \in \mathbb{R}^d : \{\mathbf{p}\} + S_2 \subseteq S_1\}$. Let $B_\gamma = \{\mathbf{p} \in \mathbb{R}^d : \|\mathbf{p}\| \leq \gamma\}$ be a ball of radius $\gamma$ centered at the origin.

Given a polytope $P \in \mathbb{R}^d$ an a real number $\gamma > 0$, let

$$P^{(+\gamma)} = P + B_\gamma,$$
$$P^{(-\gamma)} = P - B_\gamma.$$

Hence, $P^{(+\gamma)}$ and $P^{(-\gamma)}$ are the results of expanding or contracting, in a certain sense, the polytope $P$.

Also, let $P^{[+\gamma]}$ be the result of moving each halfspace defining a facet of $P$ outwards by distance $\gamma$, and similarly, let $P^{[-\gamma]}$ be the result of moving each such halfspace inwards by distance $\gamma$. Put differently, we can think of the halfspaces defining the facets of $P$ as moving outwards at unit speed, so $P$ expands with time. Then $P^{[\pm\gamma]}$ is $P$ at time $\pm\gamma$. See Figure 1.

**Observation 1.** *We have* $P^{(-\gamma)} = P^{[-\gamma]}$.

*Proof.* Each point in $P^{[-\gamma]}$ is at distance at least $\gamma$ from each hyperplane containing a facet of $P$, hence, it is at distance at least $\gamma$ from the boundary of $P$, so it is in $P^{(-\gamma)}$. Now, suppose for a contradiction that there exists a point $\mathbf{p} \in P^{(-\gamma)} \setminus P^{[-\gamma]}$. Then $\mathbf{p}$ is at distance less than $\gamma$ from a point $\mathbf{q} \in \partial h \setminus f$, where $f$ is some facet of $P$ and $\partial h$ is the hyperplane containing $f$. But then the segment $\mathbf{pq}$ must intersect another facet of $P$. $\square$

However, in the other direction we have $P^{(+\gamma)} \subsetneq P^{[+\gamma]}$. Furthermore, the Hausdorff distance between them could be arbitrarily large (see again Figure 1).

Then the $\gamma$-envelope of $P$ is given by $\partial P^{(\gamma)} = P^{(+\gamma)} \setminus P^{(-\gamma)}$, and the $\gamma$-margin of $P$ is given by $\partial P^{[\gamma]} = P^{[+\gamma]} \setminus P^{[-\gamma]}$. See Figure 2.

Since the $\gamma$-margin of $P$ is not contained in the $\gamma$-envelope of $P$, we would like to find some sufficient condition under which, for some $\gamma' < \gamma$, the $\gamma'$-margin of $P$ is contained in the $\gamma$-envelope of $P$. Our solution to this problem is given in the following theorem. Recall that $\mathcal{X}$ is the unit ball in $\mathbb{R}^d$.

**Theorem 10.** *Let $P \subset \mathbb{R}^d$ be a polytope, and let $0 < \gamma < 1$. Suppose that $P^{[-\gamma]} \cap \mathcal{X} \neq \emptyset$. Then, within $\mathcal{X}$, the $(\gamma^2/2)$-margin of $P$ is contained in the $\gamma$-envelope of $P$; meaning, $\partial P^{[\gamma^2/2]} \cap \mathcal{X} \subseteq \partial P^{(\gamma)}$.*

The proof uses the following general observation:

**Observation 2.** *Let $Q = Q(t)$ be an expanding polytope whose defining halfspaces move outwards with time, each one at its own constant speed. Let $\mathbf{p} = \mathbf{p}(t)$ be a point that moves in a straight line at constant speed. Suppose $t_1 < t_2 < t_3$ are such that $\mathbf{p}(t_1) \in Q(t_1)$ and $\mathbf{p}(t_3) \in Q(t_3)$. Then $\mathbf{p}(t_2) \in Q(t_2)$ as well.*

*Proof.* Otherwise, $\mathbf{p}$ exits one of the halfspaces and enters it again, which is impossible. $\square$

*Proof of Theorem 10.* By Observation 1, it suffices to show that $P^{[+\gamma^2/2]} \cap \mathcal{X} \subseteq P^{(+\gamma)}$. Hence, let $\mathbf{p} \in P^{[+\gamma^2/2]} \cap \mathcal{X}$ and $\mathbf{q} \in P^{[-\gamma]} \cap \mathcal{X}$. Let $s$ be the segment $\mathbf{pq}$. Let $\mathbf{r}$ be the point in $s$ that is at distance $\gamma$ from $\mathbf{p}$. Suppose for a contradiction that $\mathbf{p} \notin P^{(+\gamma)}$. Then $\mathbf{r} \notin P$. Consider $P = P(t)$ as a polytope that expands with time, as above. Let $\mathbf{z} = \mathbf{z}(t)$ be a point that moves along $s$ at constant speed, such that $\mathbf{z}(-\gamma) = \mathbf{q}$ and $\mathbf{z}(\gamma^2/2) = \mathbf{p}$. Since $\|\mathbf{r} - \mathbf{q}\| \leq 2$, the speed of $s$ is at most $2/\gamma$. Hence, between $t = 0$ and $t = \gamma^2/2$, $\mathbf{z}$ moves distance at most $\gamma$, so $\mathbf{z}(0)$ is already between $\mathbf{r}$ and $\mathbf{p}$. In other words, $\mathbf{z}$ exits $P$ and reenters it, contradicting Observation 2. $\square$

It follows immediately from Theorem 10 and Lemma 2 that the VC-dimension of the class of $t$-polytopes with envelope $\gamma$ is at most

$$\min \left\{ 2(d+1)t \log(3t), 2vt \log(3t) \right\},$$

where $v = (4/\gamma^2 + 1)^2$. Likewise, we can approximate the optimal $t$-polytope with envelope $\gamma$ by the algorithms of Theorem 7 (with parameter $\gamma' = \gamma^2/2$).

**Acknowledgments**

We thank Sasho Nikolov, Bernd Gärtner and David Eppstein for helpful discussions. L. Gottlieb and A. Kontorovich were supported in part by the Israel Science Foundation (grant No. 755/15).

## Footnotes

[1]Since the concept class of convex sets has infinite VC-dimension, without distribution assumptions, an adversarial distribution can require an arbitrarily large sample size, even in 2 dimensions [Kearns and Vazirani, 1997].

[2]Such estimates may be found in the literature for homogeneous (i.e., $b = 0$) hyperplanes (see, e.g., Bartlett and Shawe-Taylor [1999, Theorem 4.6]), but dealing with polytopes, it is important for us to allow offsets. As discussed in Hanneke and Kontorovich [2017], the standard non-homogeneous to homogeneous conversion can degrade the margin by an arbitrarily large amount, and hence the non-homogeneous case warrants an independent analysis.

[3]This does not necessary imply that *approximating* these problems requires $c^n$ operations: As hardness-of-approximation results utilize polynomial-time reductions, ETH implies only that the runtime is exponential in some polynomial in $n$.

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
