[Reviews · NeurIPS 2018]

Reviewer 1



This paper gives an algorithm for learning convex polytopes with a given margin. A convex polytope is an intersection of some number of halfspaces. The margin is defined as a notion of distance between the boundary of the polytope and the datapoints (the paper points out there are multiple definitions for this quality). The main result of this paper is to give a polynomial dependence on t, the number of halfspaces comprising the polytope in the runtime of the algorithm, which gives a super polynomial improvement over past work. These results are nearly tight, as a polynomial dependence on the margin is not possible if P \neq NP. This result is a nice improvement, and hence I am positive about it. The main downside of this paper is uses a lot of previously known results in a rather straightforward manner to achieve this. The resulting algorithm is quite simple, as is the analysis. The lower bound also follows fairly straightforwardly from previous work. However, my resulting impression remains positive. The paper is well-written and makes progress on a basic problem, so I vote to accept. comments: A key argument seems to be in line 250, where the exponential improvement on the dependence on t appears to come from. This is minor, but it appears in the title. I think you are missing an article before the word “margin”. The enumeration in line 249 takes time exponential in t. The greedy algorithm improves this — so, first, this should be emphasized and not be presented as “alternatively”. Re Lemma 4: it is a standard exercise to write dot products as a function of distances. It’s not clear this needs to be a lemma. minor: l.115 In Lemma 2, why is the second argument of the max 2(v+1)t\log(3t) and not just 2vt\log(3t)?

Reviewer 2



The paper considers the problem of PAC learning of fat convex polytopes in the realizable case. This hypothesis class is given by intersections of t fat hyperplanes, i.e., hyperplanes with margin gamma. Using standard results, the authors derive that the VC dimension of this class is quadratic in the inverse margin size and thus that the sample complexity is polylogerithmic in this quantity. As their main result, they provide two algorithms for finding with high probability a consistent fat polytope: one with exponential runtime in t and one polynomial greedy algorithm that, however, only guarantees to find a (t log n)-polytope. Complementary, the paper states two hardness of approximation results: one for finding an approximately consistent fat hyperplane, i.e., one with the minimum number of negative points on wrong side (and all positive correctly classified), and one for finding a consistent fat polytope with the minimum number of hyperplanes. Finally, the authors also show how their results relate to the alternative class of polytopes that separate points outside of their gamma-envelope (area with Euclidean distance less or equal to gamma from the polytope boundary). The topic is without a doubt highly relevant to NIPS and the results potentially of high impact. However, I do have some concerns regarding how the given results fit together to back the main claim. Ultimately this may be mostly an issue of the presentation as much could come down to a more conservative description of the main results including a more explicit statement of the learning problem (in particular the role of the parameter t). - The claimed main contribution is an algorithm for learning t-polytopes in time polynomial in t. However, the polynomial time algorithm does only guarantee to find a polytope with potentially many more sides (O(t log t) for constant epsilon and delta). In contrast, the other algorithm, which indeed finds a consistent t-polytope, is exponential in t. To what degree is it appropriate to say that either algorithm efficiently (in terms of t) learns t-polytopes? - The hardness of approximation result of Lemma 6 appears to relate to the relevant problem of finding an approximately consistent hyperplane (however, it is insufficient to conclude hardness for t larger than 1, for some of which the problem could turn out to be easier). In contrast, Theorem 7 is dealing with the problem of approximating a consistent polytope with the minimum number of sides. How does this relate to the realizable PAC learning problem? Here, we are given a dataset and know that there is a consistent t-polytope. In any case, the proof of Theorem 7 is unclear / too brief and in particular it seems Lemma 6 is not actually used in it (it might be intended as independent complementary result; the question remains, how either relates to the problem of finding a consistent t-polytope knowing that such a polytope exists). In any case, the final two pages of the paper with the main contributions are very dense and hard to follow. In contrast, the non-essential discussion of margins versus envelopes takes up a lot of space. I believe the extension to envelops would have been fine as a brief afterthought (with proof potentially delegated to supplementary material). Instead, the actual main results should be presented in a more detailed and didactically pleasing form. So I strongly recommend to shift paper real estate accordingly. Some additional minor comments: - In Lemma 2 we probably need min instead of max since we combine two valid upper bounds to the VC dimension - In the proof correct "hyperplances" - Figures 1 and 2 are very similar and probably only one is needed given the tight space limitation (see comments above) - Probably, it would be more straightforward to define a linear embedding f to be a JL transform if it satisfies the reconstruction properties (with high probability) instead of talking about "linear embeddings that satisfy the guarantees of _the_ JL transform". - In the proof of Observation 1 two times "at least" should probably be "at most" - Last line of proof of Lemma 9: "contains within _it_ a delta-net" - Theorem 10: the first part of the bound of Lemma 2 is not reflected here

Reviewer 3



NIPS review: Learning convex polytopes with margin Overview: This paper studies the problem of learning an unknown polytope P in the realizable PAC setting. The main results are upper and lower bounds on the sample and computational complexities that depend on (i) “t”, the complexity of representing P as an intersection of half-spaces and (ii) “gamma”, the margin of the input examples (an extension of the classical notion of margin w.r.t half-spaces). More specifically: 1. Upper bound [Theorem 10]: the main result is a learning algorithm whose running time (and sample complexity) exhibits polynomial dependence on t and an exponential dependence on (1/gamma) 2. Lower bound [Theorem 7]: the exponential dependence on 1/gamma is unavoidable (assuming NP\neq P). Along the way the authors also establish results in geometry. The paper is well written (for the most part). I think this paper makes a significant progress on an interesting problem and provides substantial improvements over previous works (Arriaga and Vempala [2006], and Klivans and Servedio [2008]). Comments to the authors: I did not undersatand Theorem 7: 1. Is “\tilde H” a collection of *half-spaces*? 2. Do you require that “\tilde H” partitions S the same way like “H^*” does?